# Adsorption Efficiency and Photocatalytic Activity of Silver Sulfide Nanoparticles Deposited on Carbon Nanotubes

**Gururaj M. Neelgund** [1,*], **Sanjuana Fabiola Aguilar** [1], **Erica A. Jimenez** [1] and **Ram L. Ray** [2]

1    Department of Chemistry, Prairie View A&M University, Prairie View, TX 77446, USA
2    College of Agriculture and Human Sciences, Prairie View A&M University, Prairie View, TX 77446, USA
*    Correspondence: gmneelgund@pvamu.edu

**Abstract:** A multimode, dual functional nanomaterial, CNTs-Ag$_2$S, comprised of carbon nanotubes (CNTs) and silver sulfide (Ag$_2$S) nanoparticles, was prepared through the facile hydrothermal process. Before the deposition of Ag$_2$S nanoparticles, hydrophobic CNTs were modified to become hydrophilic through refluxing with a mixture of concentrated nitric and sulfuric acids. The oxidized CNTs were employed to deposit the Ag$_2$S nanoparticles for their efficient immobilization and homogenous distribution. The CNTs-Ag$_2$S could adsorb toxic Cd(II) and completely degrade the hazardous Alizarin yellow R present in water. The adsorption efficiency of CNTs-Ag$_2$S was evaluated by estimating the Cd(II) adsorption at different concentrations and contact times. The CNTs-Ag$_2$S could adsorb Cd(II) entirely within 80 min of the contact time, while CNTs and Ag2S could not pursue it. The Cd(II) adsorption followed the pseudo-second-order, and chemisorption was the rate-determining step in the adsorption process. The Weber−Morris intraparticle pore diffusion model revealed that intraparticle diffusion was not the sole rate-controlling step in the Cd(II) adsorption. Instead, it was contributed by the boundary layer effect. In addition, CNTs-Ag$_2$S could completely degrade alizarin yellow R in water under the illumination of natural sunlight. The Langmuir-Hinshelwood (L-H) model showed that the degradation of alizarin yellow R proceeded with pseudo-first-order kinetics. Overall, CNTs-Ag$_2$S performed as an efficient adsorbent and a competent photocatalyst.

**Keywords:** CNTs; Ag$_2$S; cadmium; adsorption; alizarin yellow R





## 1. Introduction

Water pollution engrossed by heavy metals is a primary environmental, ecological, and public health concern [1]. Heavy metals are highly soluble in water and are toxic, carcinogenic, and non-degradable [2,3]. Rapid industrialization and technological development have resulted in the discharge of heavy metal-containing effluents into surface and groundwater [4]. The discharged heavy metals are adsorbed by soil and enter the human body through the food chain. Heavy metals accumulate in various organs and body tissues and can cause irreparable damage, including death [5]. Among the heavy metals, cadmium is enormously used and highly toxic [6]. The source for discharging the cadmium into the environment is industrial activities, such as electroplating, smelting, alloy manufacturing, pigments, plastic, battery, fertilizers, pesticides, pigments, dyes, textile operations, and refining [7]. In water, cadmium exists as bivalent, Cd(II), which is responsible for several adverse effects, such as kidney dysfunction, nephritis, hypertension, renal dysfunction, nervous system dysfunction, bone lesions, digestive system dysfunction, cancer, and reproductive organ damage [8–10]. The itai-itai disease, caused by Cd(II)-contaminated water from the Jinzu river in Japan, has instigated severe pain, bone fractures, proteinuria, and osteomalacia [11]. The Cd$^{2+}$ ions have a high affinity for binding to sulfhydryl (-SH) groups of proteins in biological systems [12]. The presence of Cd(II) in water can lead to severe health and environmental problems. Therefore, its elimination is critically needed. The Cd(II) present in water could be eradicated through chemical precipitation, ion exchange,

solvent extraction, membrane separation, electrochemical removal, coagulation, and adsorption [13–21]. In these techniques, adsorption is superior because of its high efficiency, relative simplicity in design, easy operation, and low operational cost [20,21]. Because of this, many adsorbents have been developed and tested for their efficiency [20–22].

Another class of pollutants that also causes major environmental problems, such as heavy metals, is azo dyes. Azo dyes have attractive colors and are enormously used in industries due to their availability and stability [23]. These dyes contain azo bonds (-N=N-) and substituted aromatic rings [24]. The complex molecular structure of azo dyes has made them recalcitrant and resistant to biodegradation [25]. Azo dyes are mutagenic, teratogenic, and carcinogenic [25]. Moreover, azo dyes can decompose into potentially carcinogenic polychlorinated naphthalenes, benzidine, and amines [26,27]. The toxicity of azo dyes can result in lung cancer, heart diseases, chromosomal aberrations, neurotoxicity, skin disease, and respiratory problems in humans [25]. These dyes can cause disorders of the central nervous system and the inactivation of enzymatic activities [28]. Beyond toxicity, azo dyes are abundantly used in industries because of their cost and advancement in colors that cover the entire spectrum. After their use in industries, about 10-15% of the azo dyes are discharged into the environment through water effluents [29,30]. Releasing azo dyes content water can result in several environmental problems including reduced light penetration in water bodies, which leads to diminished photosynthetic activities, lessened growth and reproduction of aquatic creatures, and aesthetic damage [30,31]. The consequences of the xenobiotic and recalcitrant azo dyes impact the ecosystem's structure and functioning. Considering the adverse effects, it is essential to completely obliterate azo dyes to prevent severe damage and protect health and the environment. Different techniques have been developed to remove the azo dyes in water, viz., coagulation/flocculation, ultrafiltration and membrane processing, chemical precipitation, electrochemical degradation, and ozonation [32–37]. In comparison, photocatalysis is an excellent method for eradicating azo dyes from water in an environmentally friendly approach as the end products of this process are harmless [38,39]. Due to its efficiency in degrading azo dyes, many photocatalysts have been developed to eliminate dyes [37–45]. Among azo dyes, alizarin yellow R (AYR) is a prominently used dye in industries. It is a highly water-soluble anionic dye that contains polycyclic aromatic hydrocarbons [25]. AYR is a derivative of salicylic acid and was prepared in 1887 by Rudolf Nietzki through the reaction of *m*-nitroaniline and salicylic acid [46]. AYR is an industrially important dye and is vastly used in the textile, leather, plastics, paints, and lacquer industries [24]. Furthermore, it is used as an acid-base indicator and employed in histology, stains, and nutrient media preparations [47].

Considering the health and environmental problems associated with Cd(II) and AYR, we designed the dual applicable nanocomposite, CNTs-Ag$_2$S for efficient Cd(II) adsorption and photodegradation of AYR. The multimode CNTs-Ag$_2$S was produced through the facile deposition process of Ag$_2$S nanoparticles over oxidized carbon nanotubes (CNTs) using the hydrothermal method. The adsorption efficiency of CNTs-Ag$_2$S was estimated by evaluating the adsorption rate of Cd(II) from water. The dynamics and controlling mechanisms of Cd(II) adsorption were assessed by pseudo-first- and second-order kinetic models. The reaction pathways and the rate-controlling step underlying Cd(II) adsorption were evaluated using the Weber−Morris intraparticle pore diffusion model. The adsorption equilibrium was determined by fitting the experimental results with Langmuir, Freundlich, and Temkin isotherm models. Furthermore, the catalytic activity of CNTs-Ag$_2$S was determined through the degradation of AYR under exposure to natural sunlight. The photocatalytic activity of CNTs-Ag$_2$S was quantified using the Langmuir-Hinshelwood (L-H) model.

## 2. Results and Discussion

The ATR-FTIR spectrum of CNTs-COOH, shown in Figure 1a, demonstrated a band at 3427 cm$^{-1}$ corresponding to the O-H bond. The band for the C=O bond of the -COOH groups appeared at 1699 cm$^{-1}$, and the band for the C=C bonds of CNTs was found at

1554 cm$^{-1}$ [48–50]. The spectrum of Ag$_2$S (Figure 1b) displayed the peak of the Ag-S bond at 552 cm$^{-1}$ [51]. The spectrum of CNTs-Ag$_2$S (Figure 1c) revealed the characteristic absorption bands related to CNTs and Ag$_2$S. The prominent peaks observed at 1066 and 1096 cm$^{-1}$ in Figure 1c were attributed to C-O stretching. The peak at 3574 cm$^{-1}$, was due to the −OH stretching vibrations of adsorbed water molecules. The band due to C=O was observed at 1693 cm$^{-1}$, and the peak, at 1517 cm$^{-1}$, was attributed to C-H. The peak related to Ag-S was situated around 550 cm$^{-1}$. The XRD pattern of CNTs-Ag$_2$S (Figure 2) showed a characteristic (0 0 2) reflection of hexagonal graphite of CNTs at 26.1° [52]. It revealed the reflections related to Ag$_2$S at 22.7 (−1 0 1), 25.2 (−1 1 1), 26.5 (0 1 2), 29.2 (1 1 1), 31.7 (−1 1 2), 33.8 (1 2 0), 34.6 (−1 2 1), 34.9 (0 2 2), 36.8 (1 1 1), 37.0 (1 2 1), 37.3 (0 1 3), 37.9 (−1 0 3), 40.9 (0 3 1), 43.6 (0 2 3), 44.4 (−1 3 1), 46.4 (−1 2 3), 48.0 (−2 1 2), 48.9 (0 1 4), 53.0 (0 4 0), 53.5 (−2 1 3), 58.3 (−1 4 1), 58.4 (−2 2 3), 60.2 (−1 0 5), 61.4 (0 1 5), 62.8 (2 3 1), 63.4 (2 1 3), 63.9 (−1, 3 4), and 68.0° (2 3 2). These values agree with the values found in the standard pattern of acanthite Ag$_2$S (JCPDS file 14-0072) [52]. The TEM images of CNTs-Ag$_2$S presented in Figure 3 explored the deposition of spherical-resembling Ag$_2$S nanoparticles over the surface of tubular-structured CNTs. The proportion of Ag$_2$S nanoparticles was low, possibly due to the low quantity of precursor AgNO$_3$ used in the preparation compared to the ratio of CNTs. The presence of Ag$_2$S nanoparticles over the surface of CNTs is perceptible in Figure 3a–f. It is noticeable in Figure 3a that the few Ag$_2$S nanoparticles have combined and formed clusters. Figure 3b,d reveal the ruptured surface of some CNTs. It could have happened through the harsh treatment of CNTs with the mixture of concentrated HNO$_3$ and H$_2$SO$_4$. This process was required to make the CNTs hydrophilic and provide a suitable environment for the effective deposition of Ag$_2$S nanoparticles. However, the entire surface of the CNTs was not distracted and the smooth surface of the CNT is perceptible in Figure 3e,f. Overall, a few defective sites were formed in CNTs. Figure 3d-f reveals the strong adherence of Ag$_2$S nanoparticles to the smooth-surfaced CNT. The absence of leached or free-standing nanoparticles in the void area shows the strong adherence of Ag$_2$S nanoparticles to the surface of CNTs. The size of the Ag$_2$S nanoparticle present in Figure 3d,e, was around 38 and 15 nm, respectively. Therefore, Ag$_2$S nanoparticles have broad size distribution. The average size of Ag$_2$S nanoparticles, calculated from XRD, was around 29 nm, which agrees with the size determined from the TEM images. The core of the Ag$_2$S nanoparticles (Figure 3d) appears denser than the edge. The CNTs have a width of around 110 nm and a length of several micrometers. Overall, CNTs have effectively exfoliated. Further, the EDS of CNTs-Ag$_2$S (Figure S1, Supplementary Materials) demonstrated the presence of O, Ag, and S in CNTs-Ag$_2$S. The peaks for Cu and Ca were found in Figure S1, those are by the copper grid used in the TEM measurement.

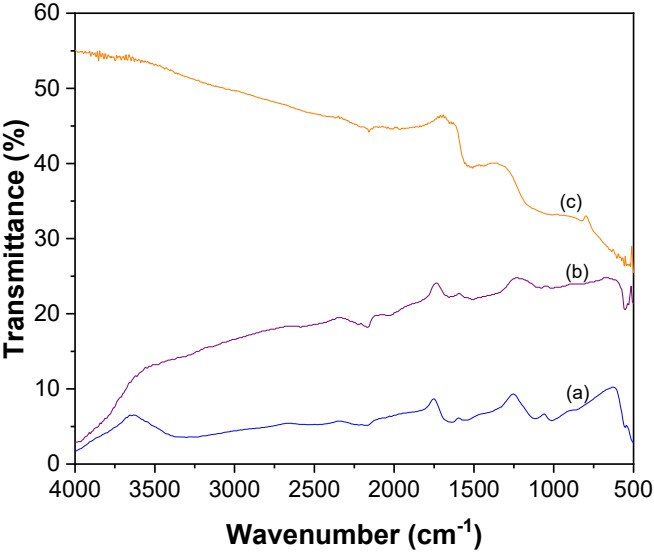

**Figure 1.** ATR-FTIR spectra of (**a**) CNTs-COOH, (**b**) Ag$_2$S, and (**c**) CNTs-Ag$_2$S.

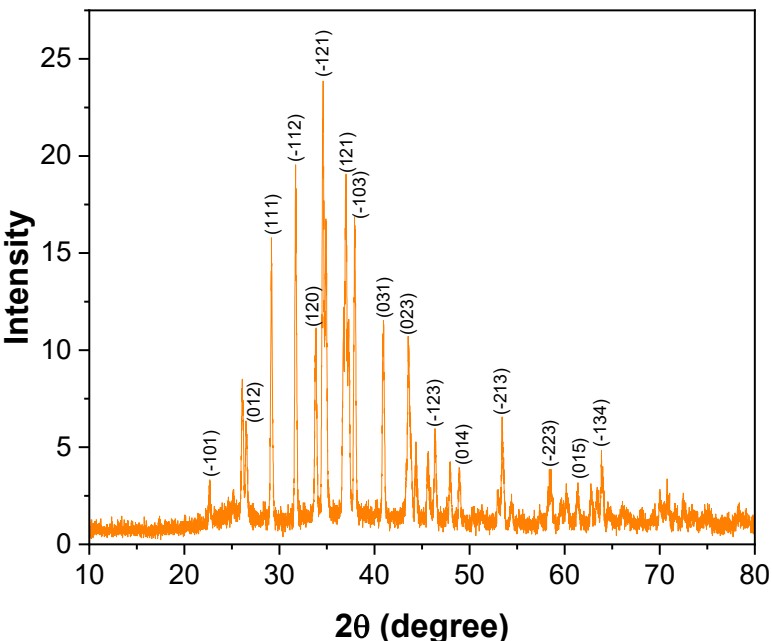

**Figure 2.** XRD of CNTs-Ag$_2$S.

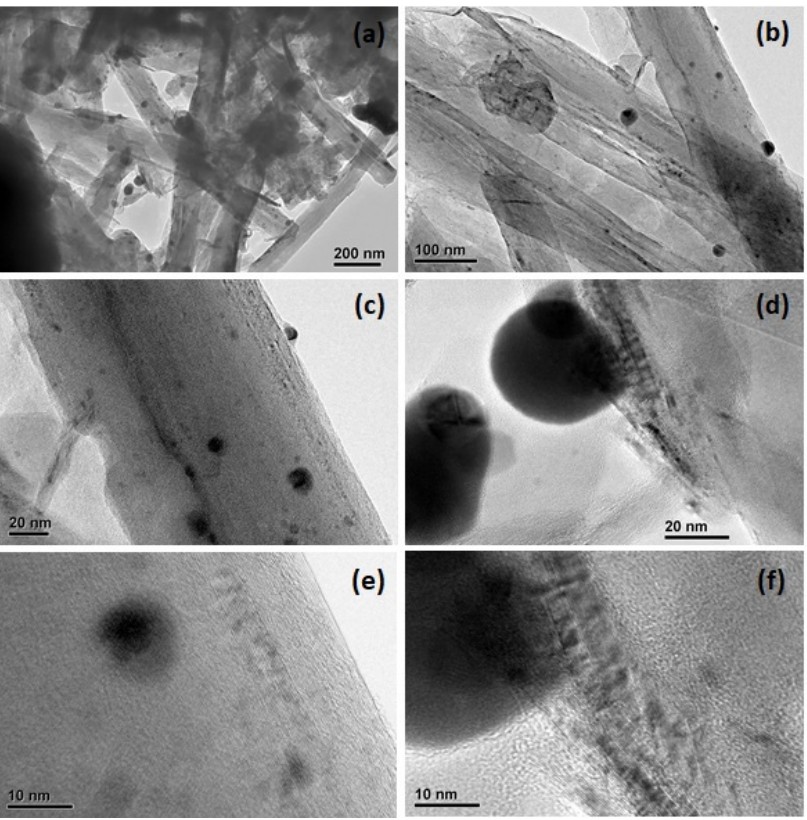

**Figure 3.** (**a**–**f**) TEM images of CNTs-Ag$_2$S.

The TGA pattern of CNTs-Ag$_2$S (Figure 4) displayed four weight loss steps. The initial weight loss of 6.5% that occurred before 100 °C was attributed to the desorption of physically adsorbed water molecules over the surface of CNTs-Ag$_2$S. The following weight loss of 15% ensued within the range of 100–490 °C was due to the detachment and decomposition of oxygen-containing functional groups existing on the surface of the CNTs. The subsequent weight loss of 10.5% transpired within the range of 500–540 °C, was owing

to the breaking of the bond between silver and sulfide in Ag₂S nanoparticles and releasing sulfur. The successive sharp and significant weight loss of 31.5% taken place within the range of 540–550 °C was by the decomposition of sulfur and the formation of metallic silver and silver oxide. The residual weight remained was about 64%. The broad endothermic peak found in the DTA curve around 80 °C was because of the dehydration of the sample. The small endothermic peak found at 170 °C accounted for the removal of functional groups from the surface of the CNTs. The intense exothermic peak at 545 °C was assigned to release sulfur by breaking the bond between silver and sulfide. The UV-vis absorption spectrum of CNTs-Ag₂S, shown in Figure 5, exhibited a characteristic absorption band of the C=C bonds of CNTs at 232 nm [52]. In addition, the broad absorption tail corresponding to Ag₂S nanoparticles was observed in the visible region [52,53]. If semiconductor nanoparticles are conjugated to CNTs, it shows the charge-transfer band [52,54]. However, no such band was observed in Figure 5, illustrating that the conjugation of Ag₂S nanoparticles to CNTs has not altered their energy states [52,54].

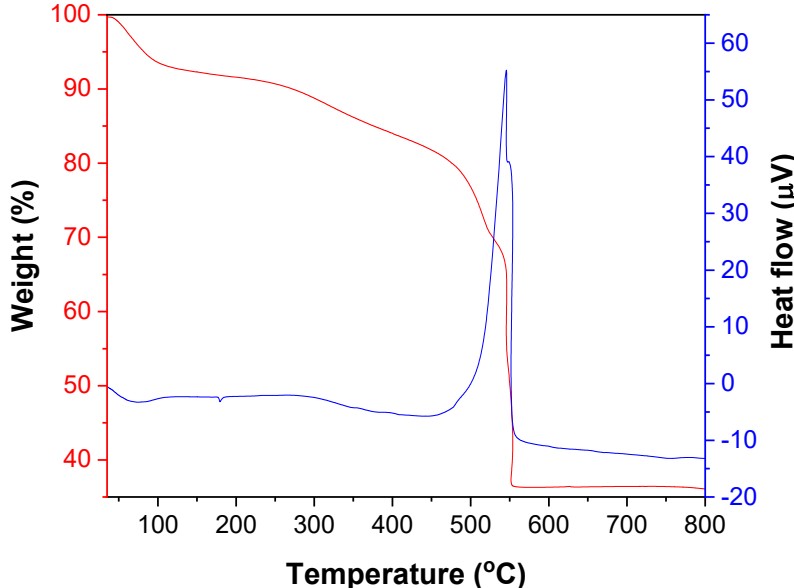

**Figure 4.** TG/DTA of CNTs-Ag₂S.

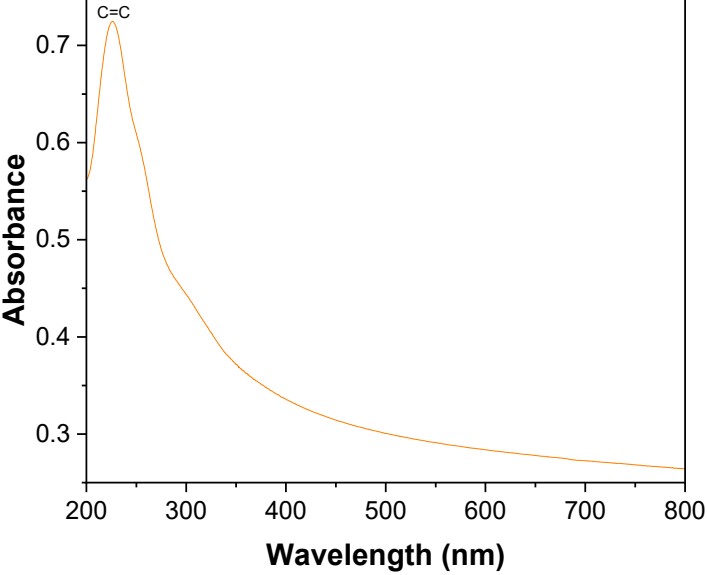

**Figure 5.** UV-vis absorption spectrum of CNTs-Ag₂S.

The XPS survey spectrum of CNTs-Ag$_2$S, shown in Figure 6a, confirmed the presence of C, O, Ag, and S. The high-resolution spectrum of C1s (Figure 6b) divulged four distinct peaks by Gaussian fitting situated at 284.5, 285.5, 286.4, and 287.8 eV. Among these, peaks at 284.5 and 285.5 eV were assigned to the C-C bonds of the sp$^2$ and sp$^3$ hybridized carbon atoms of CNTs, respectively [55]. The peaks at 286.4 and 287.8 eV were due to the C-O and C=O bonds of CNTs, respectively [55]. The deconvoluted O 1s peak (Figure 6c) exhibited a peak at 531.3 eV for the lattice oxygen and a peak at 532.4 eV corresponding to the carbonyl (=C-O) functional groups of the CNTs [56]. The core-electron binding energy of Ag 3d3/2 was found at 373.9 eV, and that of Ag 3d5/2 was at 367.9 eV (Figure 6d). The presence of Ag 3d3/2 and Ag 3d5/2 peaks and their positions confirm the Ag$^+$ state of silver [57]. The difference in the position of Ag 3d5/2 and Ag 3d3/2 peaks was 6 eV, and no shoulder or satellite peaks were observed between them. The high-resolution S 2p spectrum (Figure 6e) for the spin-orbit splitting of S$^{2+}$ was deconvoluted into the 2p3/2 level peak at 161.2 eV, and the 2p1/2 peak at 162.3 eV. These peaks occurred by the Ag-S bonds of the Ag$_2$S nanoparticles [58]. The additional peak at 163.9 eV is related to the presence of sulfur in CNTs-Ag$_2$S [58].

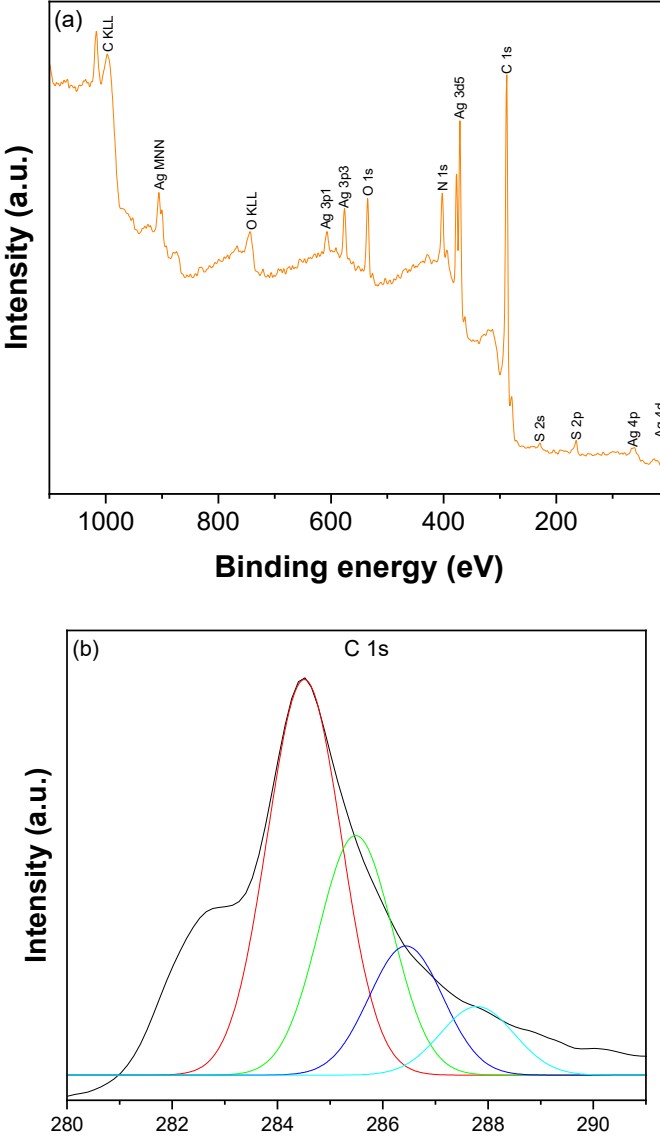

**Figure 6.** *Cont.*

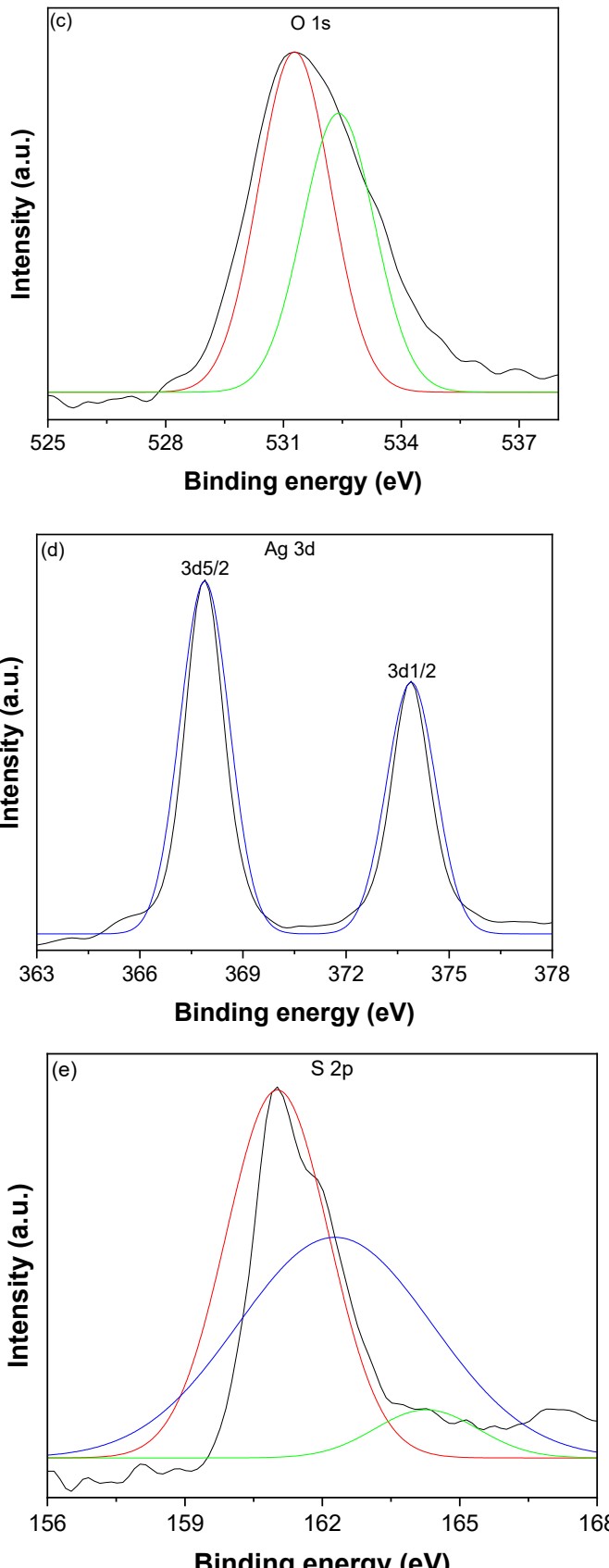

**Figure 6.** (**a**) XPS survey spectrum of CNTs-Ag$_2$S. (**b**) High-resolution spectrum of C1s. (**c**) High-resolution spectrum of O1s. (**d**) High-resolution spectrum of Ag3d. (**e**) High-resolution spectrum of S2p.

The adsorption ability of CNTs-Ag$_2$S was determined by estimating the Cd(II) adsorption present in water. The contact time between adsorbent and adsorbate is critical and it controls adsorption. Therefore, the Cd(II) adsorption was determined by allowing for contact with CNTs-Ag$_2$S at different times and compared with that of CNTs and Ag$_2$S (Figure 7). The plot of the adsorption capacity, q$_t$ versus t, is presented in Figure S2. Cd(II) adsorption is time dependent and occurs as a gradient function of time. Hence, contact time is crucial in controlling the Cd(II) adsorption. The tendency in Cd(II) adsorption by CNTs, Ag$_2$S, and CNTs-Ag$_2$S was identical like the adsorption was rapid, then gradually reduced, and finally attained equilibrium. The initial rapid Cd(II) adsorption could have occurred due to the difference in the concentration of Cd$^{2+}$ ions in the solution and over the surface of the CNTs-Ag$_2$S, which facilitated the movement of Cd$^{2+}$ ions from the solution to the surface of the CNTs-Ag$_2$S [8]. Also, in the beginning, many active sites were available on the surface of CNTs-Ag$_2$S to occupy by Cd$^{2+}$ ions [8]. With the elapsed contact time, the active sites of the CNTs-Ag$_2$S were predominantly occupied by Cd$^{2+}$ ions. Apart, there is a possibility of repulsion between Cd$^{2+}$ ions in the solution and Cd$^{2+}$ ions exist over the surface of CNTs-Ag$_2$S. Owing to these reasons, the adsorption rate could gradually decrease and eventually attain equilibrium [8]. Within 80 min of the contact time, CNTs-Ag$_2$S could adsorb the Cd(II) completely; however, under identical conditions, the adsorption rates of CNTs and Ag$_2$S were 79 and 53%, respectively. Thus, the adsorption efficiency of CNTs and Ag$_2$S was significantly improved by their conjugation in CNTs-Ag$_2$S. Further, to identify the adsorption kinetics of CNTs-Ag$_2$S, the data were simulated with the pseudo-first-order kinetic model using the expression presented in Equation (1).

$$ln(q_e - q_t) = \ lnq_e - k_1t \tag{1}$$

where $q_e$ and $q_t$ are the quantity of adsorbate (mg/g) at equilibrium and particular time, $t$ (min), respectively, and $k_1$ (min$^{-1}$) is the pseudo-first-order rate constant.

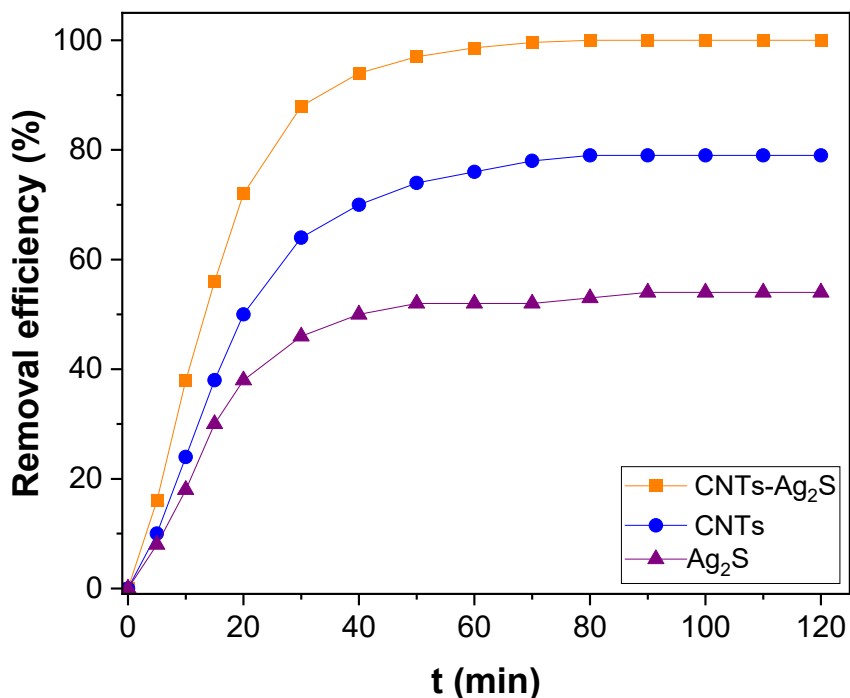

**Figure 7.** Adsorption kinetics of Cd(II) over CNTs, Ag$_2$S, and CNTs-Ag$_2$S.

The pseudo-first-order plot obtained for Cd(II) adsorption over CNTs, Ag$_2$S, and CNTs-Ag$_2$S is shown in Figure 8. The correlation coefficients (R$^2$) perceived for CNTs, Ag$_2$S, and CNTs-Ag$_2$S were 0.9906, 0.9540, and 0.9930, respectively. The k$_1$ and q$_e$ were computed

using the slope and intercept values of the straight lines acquired in Figure 8. The estimated values of $k_1$ and $q_e$ is summarized in Table 1. The calculated $q_e$(cal) could not agree with the experimental value $q_e$(exp) for all samples. Thus, the experimental data for the Cd(II) adsorption could not be projected by the pseudo-first-order kinetic model. Alternatively, it was analyzed using the pseudo-second-order kinetic model using Equation (2).

$$\frac{t}{q_t} = \frac{1}{k_2\,q_e^2} + \frac{t}{q_e} \tag{2}$$

where $k_2$ [g/(mg.min)] is the pseudo-second-order rate constant.

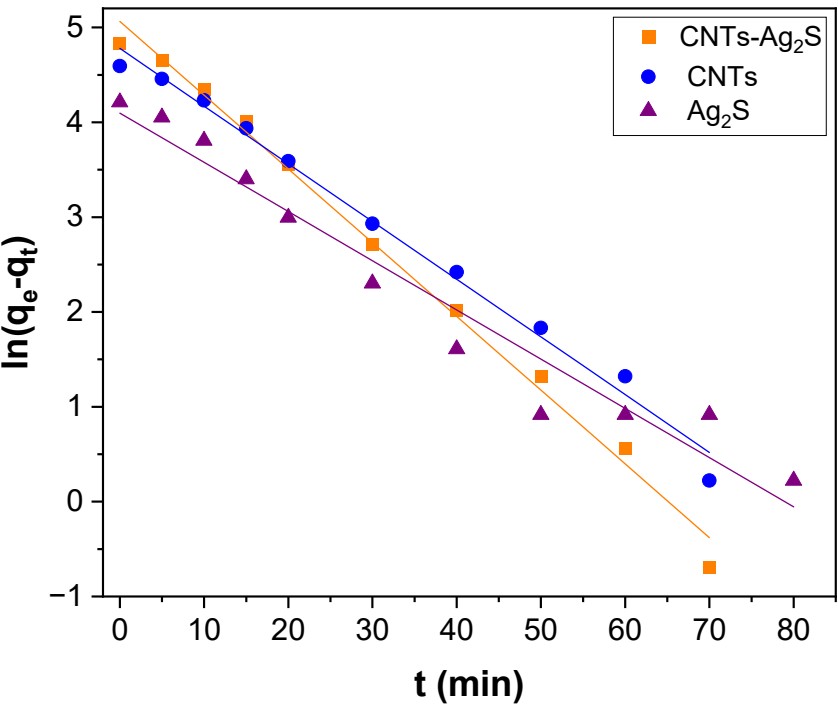

**Figure 8.** Pseudo-first-order kinetics for Cd(II) adsorption over CNTs, Ag$_2$S, and CNTs-Ag$_2$S.

**Table 1.** Parameters calculated for the Cd(II) adsorption using adsorption kinetic models.

| Adsorbent | $q_e$ (exp) mg g$^{-1}$ | Pseudo-First-Order Kinetic Model | | | Pseudo-Second-Order Kinetic Model | | |
|---|---|---|---|---|---|---|---|
| | | $q_e$ (cal) mg g$^{-1}$ | $k_1$(min$^{-1}$) | R$^2$ | $q_e$ (cal) mg g$^{-1}$ | $k_2 \times 10^{-3}$ (g mg$^{-1}$ min$^{-1}$) | R$^2$ |
| CNTs-Ag$_2$S | 125.0 | 5.0633 | 0.0778 | 0.9930 | 133.2 | 1.2140 | 0.9980 |
| CNTs | 98.75 | 4.7785 | 0.0608 | 0.9906 | 109.8 | 0.8360 | 0.9963 |
| Ag$_2$S | 67.50 | 4.0970 | 0.0519 | 0.9540 | 72.62 | 1.7750 | 0.9985 |

The pseudo-second-order model plot is illustrated in Figure 9, and the determined parameters are presented in Table 1. The R$^2$ values derived from the pseudo-second-order plots for CNTs, Ag$_2$S, and CNTs-Ag$_2$S were 0.9963, 0.9985, and 0.9980, respectively. The R$^2$ value of the second-order plot was relatively higher than that found for the first-order plot, which was close to 1. The $q_e$(cal) of the second-order model matched the $q_e$(exp) for all samples. The linearity of the pseudo-second-order plot and the agreeing values of $q_e$(exp) and $q_e$(cal) show that the Cd(II) adsorption could be analyzed using the pseudo-second-order kinetics rather than the pseudo-first-order kinetics. The agreement of the second-order kinetics demonstrates that chemisorption was the rate-determining step in the Cd(II) adsorption [9]. Moreover, it represents the rapid transfer of Cd(II) from a solution with a lower initial concentration to the surface of the CNTs-Ag$_2$S due to

the concentration gradient [9]. Further, the experimental data were explored using the Weber−Morris intraparticle pore diffusion model to evaluate the diffusion mechanism using Equation (3).

$$q_t = k_{id}\, t^{0.5} + C \tag{3}$$

where $k_{id}$ (mg/g.min) is the intraparticle diffusion rate constant, which can be calculated from the linear plot of $q_t$ versus $t^{0.5}$. c (mg/g) is the intraparticle diffusion constant, estimated from the intercept and directly proportional to the boundary layer thickness. It is assumed that the higher the intercept value, the more significant the contribution of the surface adsorption in the rate-controlling step. If the regression of the $q_t$ versus $t^{0.5}$ plot is linear and passes through the origin, intraparticle diffusion plays a significant role in controlling the kinetics of the adsorption process. If it does not pass through the origin, intraparticle diffusion is not the only rate-limiting step. Instead, it is contributed by the boundary layer effect [59]. The intraparticle diffusion plot of $q_t$ versus $t^{0.5}$ acquired for the Cd(II) adsorption on CNTs-Ag$_2$S is demonstrated in Figure 10. The plot in Figure 10 does not pass through the origin of the coordinates (Figure S3). So, intraparticle diffusion is not the sole rate-limiting step in Cd(II) adsorption over CNTs-Ag$_2$S. The association of the two straight lines in the intraparticle diffusion plot (Figure S3) and the intercept values (Table S1) support that the intraparticle diffusion was not only a rate-controlling step in the Cd(II) adsorption; it also contributed through the boundary layer effect [59]. The multilinearity of the plot in Figure 10 (Figure S3) shows that the Cd(II) adsorption transpires through multiple phases [60]. Among these, the initial stage occurred through the rapid adsorption of Cd$^{2+}$ ions, the second phase was owing to the diffusion of Cd$^{2+}$ ions into the pores of CNTs-Ag$_2$S, and the third phase was due to equilibrium of the adsorption that caused a chemical reaction/bonding [60].

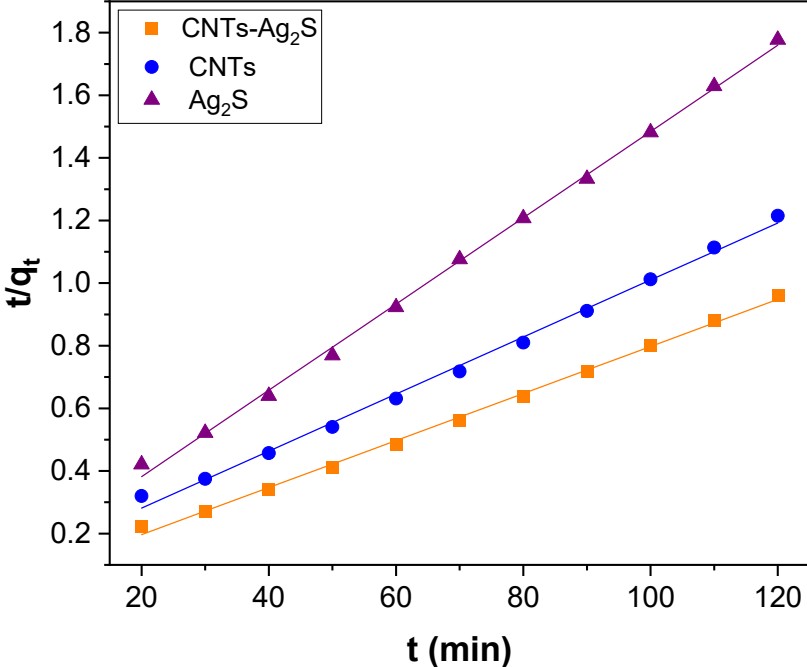

**Figure 9.** Pseudo-second-order kinetics for Cd(II) adsorption over CNTs, Ag$_2$S, and CNTs-Ag$_2$S.

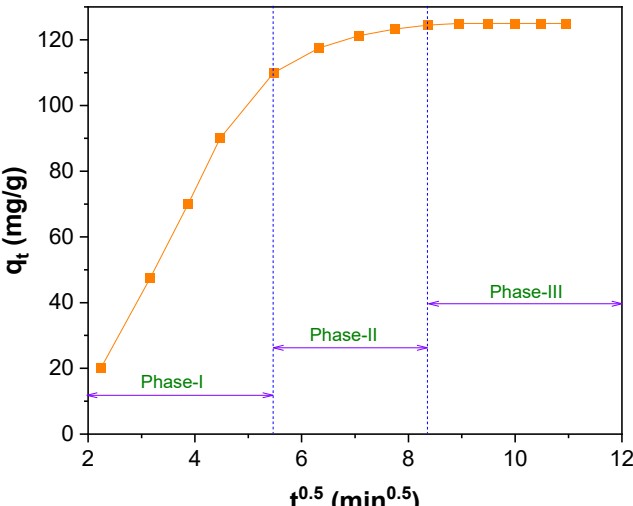

**Figure 10.** Weber-Morris intraparticle diffusion plot for the Cd(II) adsorption over CNTs-Ag$_2$S.

The experimental equilibrium parameters for Cd(II) adsorption were determined by applying three isotherm models: Langmuir, Freundlich, and Temkin. The Langmuir model predicts that the adsorbed molecules form a monolayer and adsorption emerges at a static number of adsorption sites, each of which is equivalent in efficiency. Moreover, each molecule has a constant enthalpy and adsorption activation energy, it means that all molecules have an affinity equal to entire adsorption sites [61]. With the help of Langmuir isotherm model, it could be find the value of the maximum adsorption capacity of the adsorbent using Equation (4) [62]:

$$\frac{C_e}{q_e} = \frac{C_e}{q_m} + \frac{1}{K_L\,q_m} \tag{4}$$

where $q_e$ (mg/g) is the amount of adsorbed Cd(II) per unit mass of CNTs-Ag$_2$S; $C_e$ (mg/L) is the concentration of Cd(II) at equilibrium; $q_m$ is the maximum amount of the Cd(II) adsorbed per unit mass of CNTs-Ag$_2$S to form a complete monolayer on the surface-bound at high $C_e$. $K_L$ is the Langmuir adsorption constant related to the free energy of adsorption. The Langmuir plot for Cd(II) adsorption over CNTs-Ag$_2$S is presented in Figure 11. The estimated parameters are shown in Table 2. The maximum adsorption capacity (q$_m$) calculated for Cd(II) adsorption over CNTs-Ag$_2$S was 256.4 mg/g.

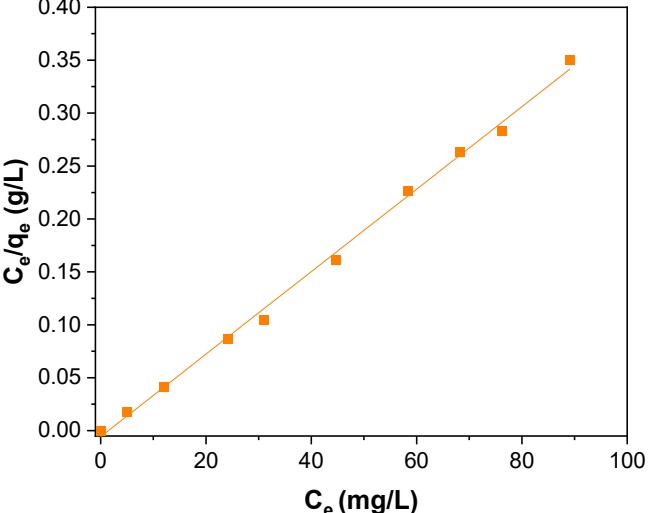

**Figure 11.** Langmuir isotherm plot for Cd(II) adsorption over CNTs-Ag$_2$S.

**Table 2.** Parameters calculated for Cd(II) adsorption over CNTs-Ag$_2$S using Langmuir, Freundlich, and Temkin adsorption isotherm models.

| Langmuir Isotherm | | | Freundlich Isotherm | | | Temkin Isotherm | | |
|---|---|---|---|---|---|---|---|---|
| q$_m$ (mg g$^{-1}$) | K$_L$ (L mg$^{-1}$) | R$^2_{Lan}$ | K$_F$ (mg g$^{-1}$) | n | R$^2_{Fre}$ | A | B | R$^2_{Tem}$ |
| 256.4 | −0.6830 | 0.9972 | 305.0 | −31.21 | 0.3562 | $1.1 \times 10^{256}$ | 0.4565 | 0.0019 |

The Freundlich isotherm is developed based on the assumption that the adsorption sites are distributed exponentially concerning the heat of adsorption [62,63]. It provides the relationship between the equilibrium of liquid and solid phase capacities with multi-layer adsorption properties consisting of the heterogeneous surface of the adsorbent. The Freundlich isotherm that supports multilayer adsorption agrees with the Langmuir model over moderate ranges of concentrations but differs at low and high concentrations. The linear form of the Freundlich isotherm can be represented b Equation (5):

$$ln\ q_e = ln\ K_F + \frac{ln\ C_e}{n} \tag{5}$$

where $q_e$ (mg/g) is the amount of Cd(II) adsorbed at equilibrium. $K_F$ and n are Freundlich constants. $K_F$ symbolizes the affinity of the adsorbent, and n signifies the adsorption intensity. $C_e$ (mg/L) is the Cd(II) concentration at equilibrium. The Freundlich isotherm plot attained for Cd(II) adsorption is shown in Figure S4, and the calculated parameters are included in Table 2.

The Temkin isotherm model is based on the fact that, during adsorption, the heat of all molecules decreases linearly with an increase in the coverage of the adsorbent surface and that adsorption is characterized by the uniform distribution of binding energies up to the maximum binding energy [62,64]. The linear form of the Temkin isotherm model is shown in Equation (6):

$$q_e = B\ lnA + B\ ln\ C_e \tag{6}$$

where B = RT/K$_T$, K$_T$ is the Temkin constant related to the heat of adsorption (J/mol); A is the Temkin isotherm constant (L/g), R is the gas constant (8.314 J/mol K), and T is the absolute temperature (K). The Temkin isotherm fitting plot for the Cd(II) adsorption is shown in Figure S5. The values estimated from Figure S5 are illustrated in Table 2.

The R$^2_{Lan}$ value found for the Langmuir isotherm plot was 0.9972; for Freundlich and Temkin isotherm plots, the R$^2_{Fre}$ and R$^2_{Tem}$ values were 0.3562 and 0.0019, respectively. The Freundlich and Temkin isotherm models provided scattered points and failed to accomplish linearity. Therefore, the Freundlich and Temkin isotherm models were unsuitable for explaining Cd(II) adsorption over CNTs-Ag$_2$S. However, the Langmuir isotherm model produced linearity. Therefore, the Langmuir model is appropriate for elucidating the Cd(II) adsorption. The validation of the Langmuir model indicates that the Cd(II) adsorption over CNTs-Ag$_2$S ensued with monolayer molecular covering and chemisorption, together [65].

Further, the photocatalytic activity of CNTs-Ag$_2$S was investigated through the degradation of AYR under exposure to sunlight. The degradation of AYR by CNTs-Ag$_2$S is presented in Figure 12. The activity of CNTs-Ag$_2$S was compared with that of CNTs and Ag$_2$S. The CNTs-Ag$_2$S could degrade the AYR completely within 120 min of illumination. However, CNTs and Ag$_2$S were capable to degrade 77.3 and 41% of AYR, respectively, in 120 min. Hence, the conjugation of CNTs and Ag$_2$S was substantially improved the photocatalytic activity in CNTs-Ag$_2$S. The degradation of AYR was quantified by the reduction in the electronic absorption band located at 373 nm (Figure 13). The hybrid nano-architecture of CNTs-Ag$_2$S was proficient in adsorbing higher number of AYR molecules and capable in absorption of sunlight. In addition, the effectual hindrance of recombining of electrons and holes during photocatalysis facilitated the degradation process. The photocatalytic activity of CNTs-Ag$_2$S was further explored by finding the apparent rate constants using

the Langmuir-Hinshelwood (L-H) model and with the help of Equation (7) that could be applicable for low concentrations of dyes [38,39,44,45].

$$\ln \frac{C_0}{C} = k_{app}\, t \qquad (7)$$

where $C_0$ is the initial concentration of AYR and $C$ is the concentration at a particular time of irradiation. $k_{app}$ is the apparent rate constant of the reaction, and t is the irradiation time.

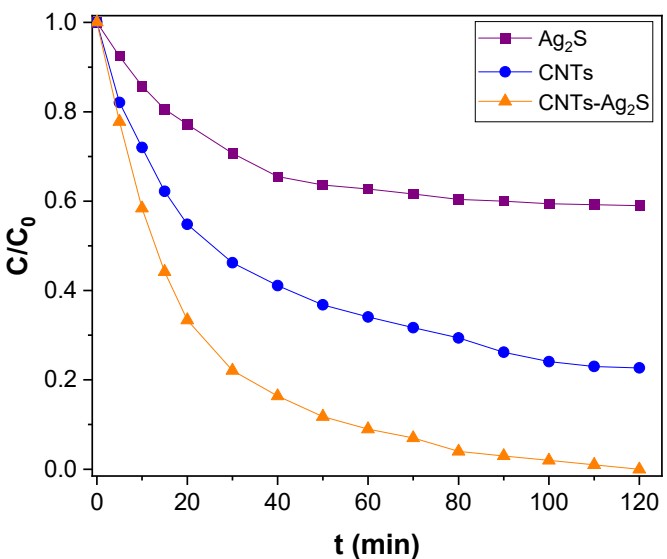

**Figure 12.** The degradation of alizarin yellow R in the presence of CNTs, Ag$_2$S, and CNTs-Ag$_2$S under the illumination to sunlight.

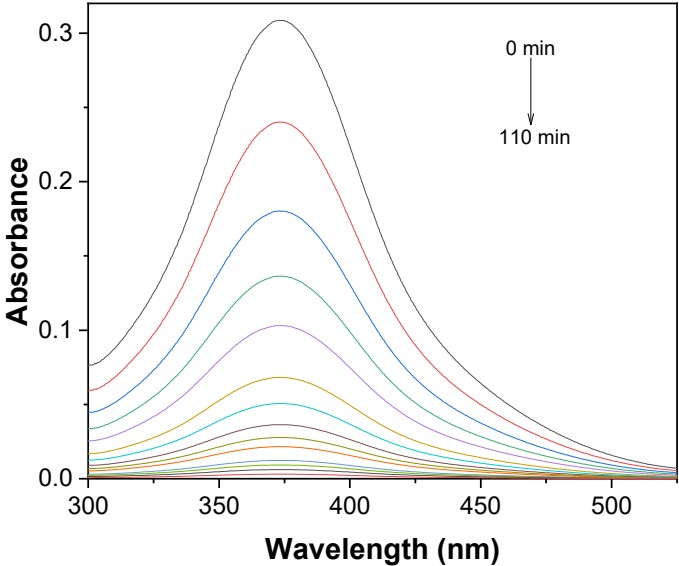

**Figure 13.** Reduction in the electronic absorption of alizarin yellow R in the presence of CNTs-Ag$_2$S under the irradiation to sunlight.

The L-H plots perceived for the degradation of AYR were linear, as depicted in Figure 14. The linearity of the L-H plots reveals that the degradation of AYR ensue with pseudo-first-order kinetics. The apparent rate constants determined by the L-H plots for CNTs, Ag$_2$S, and CNTs-Ag$_2$S were 0.0108, 0.0035, and 0.0378 min$^{-1}$, respectively. The value determined for CNTs-Ag$_2$S was about four-fold higher than CNTs and eleven-fold greater

than Ag$_2$S. Therefore, the conjugation of CNTs and Ag2S has magnificently improved the photocatalytic activity in CNTs-Ag$_2$S. Due to the conjugated structure, containing aromatic, carbonyl, and azo groups, the AYR solution possessed an intense color in water. Accordingly, AYR degradation could happen by breaking the conjugated system in the presence of CNTs-Ag$_2$S under sunlight irradiation. With this assumption, the possible mechanism for the rapid degradation of AYR by CNTs-Ag$_2$S could be explained as follows (Figure 15).

$$CNTs\text{-}Ag_2S + h\nu \rightarrow e^- + h^+$$
$$h^+ + H_2O \rightarrow HO^- + H^+$$
$$e^- + O_2 \rightarrow O_2{}^{\bullet-}$$
$$O_2{}^{\bullet-} + H^+ \rightarrow HO_2{}^{\bullet}$$
$$HO_2{}^{\bullet} + H_2O \rightarrow H_2O_2 + HO^{\bullet}$$
$$AYR + HO^{\bullet} \rightarrow H_2O + CO_2 + \text{nontoxic products}$$

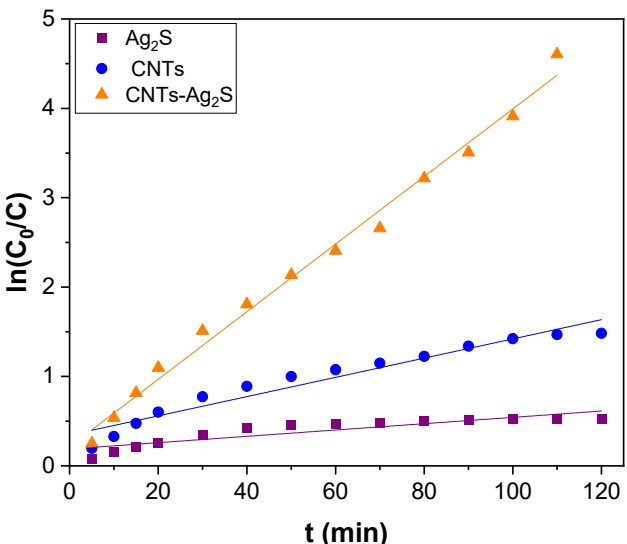

**Figure 14.** Langmuir-Hinshelwood plot for the degradation of alizarin yellow R in the presence of CNTs-Ag$_2$S under illumination to sunlight.

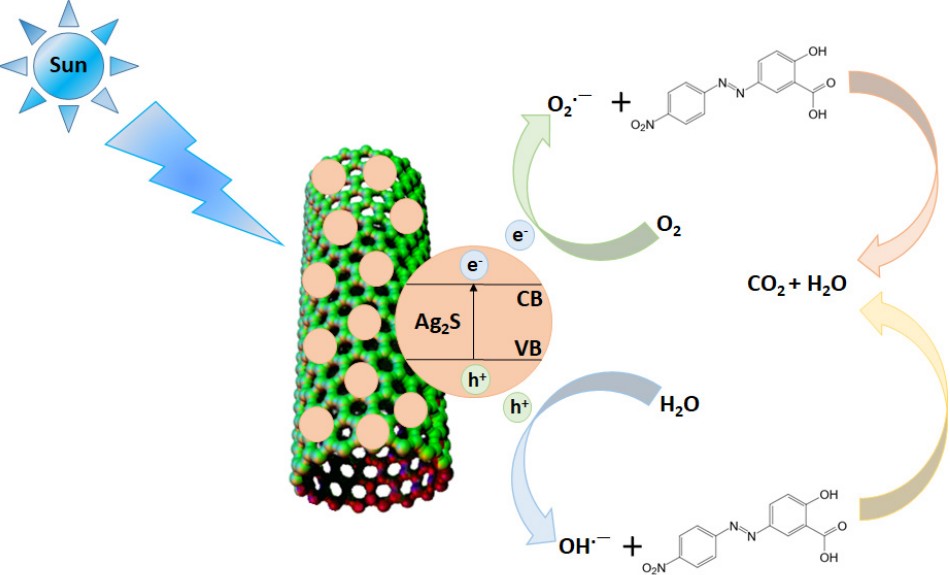

**Figure 15.** A possible mechanism for the degradation of alizarin yellow R in the presence of CNTs-Ag$_2$S under exposure to sunlight.

Further, to find the stability of CNTs-Ag$_2$S, it was recovered by centrifugation after photocatalysis. It was washed with DI water, dried and used in further two cycles of the photocatalysis. The reused CNTs-Ag$_2$S was able to degrade AYR completely in further two cycles without any significant reduction in activity (Figure S6). The XRD of CNTs-Ag$_2$S recorded before and after using in the three degradation cycles of AYR could not show any structural modification or rapture (Figure S7). Consequently, CNTs-Ag$_2$S is table and suitable for reuse.

The CNT/Ag$_2$S nanocomposite prepared by Di et al. and applied it to the degradation of rhodamine B revealed its higher activity than bare Ag$_2$S nanoparticles under illumination to visible and near-infrared (NIR) light [57]. In addition, the recycled CNT/Ag$_2$S nanocomposite could not lost the activity [57]. The Ag$_2$S-CNT nanocomposite, reported by Meng et al. [66], efficiently degraded texbrite BA-L in presence of visible light. In this study, the Ag$_2$S-CNT nanocomposite was for four degradation cycles of texbrite BA-L [66]. Further, the photocatalytic activity of CNTs-Ag$_2$S in the degradation of AYR was compared with different photocatalysts and presented in Table 3 [24,25,46,67–69].

**Table 3.** Comparison of the degradation rate of AYR by CNTs-Ag$_2$S with reported photocatalysts.

| Photocatalyst | Source of Irradiation | Degradation Rate (%) | Ref |
|---|---|---|---|
| CNTs-Ag$_2$S | Sunlight | 100 | This work |
| Fe nanoparticles | Sunlight | 93.7 | 25 |
| H$_2$O$_2$ | UV | 100 | 24 |
| Zinc oxide | UV | 92.5 | 46 |
| β-MnO$_2$ nanowires | Mercury lamp | 98.0 | 69 |
| Mn$_3$O$_4$ | Mercury lamp | 62.0 | 69 |
| MnO(OH) nanorods | Mercury lamp | 54.0 | 69 |
| Bi$_2$O$_3$@RGO | Sunlight | 41.5 | 70 |
| ZnO nanoparticles | UV | 95.0 | 71 |
| ZnO nanoparticles | Sunlight | 13.2 | 71 |
| ZnO nanoparticles | Visible | 06.2 | 71 |

## 3. Experimental

### 3.1. Materials

The chemicals and CNTs were purchased from Millipore Sigma and used as received. The aqueous solutions were prepared using ultrapure water obtained by the Milli-Q Plus system (Millipore; Burlington, MA, USA).

### 3.2. Preparation of CNTs-Ag$_2$S

The hydrophobic pristine CNTs were modified to hydrophilic through the oxidization by refluxing in a mixture of 1:3 (*v/v*) concentrated HNO$_3$ and H$_2$SO$_4$ at 70 °C for 24 h. The resulting oxidized CNTs were collected through centrifugation and purified by washing them with DI water [70,71]. The resulting oxidized CNTs were dried under a vacuum and used to deposit Ag$_2$S nanoparticles. About 40 mg of oxidized CNTs (CNTs-COOH) were dispersed in 40 mL of DI water using sonicator, and a solution of 0.04 mol/L of AgNO$_3$ in 40 mL of DI water was added. The mixture was allowed to stir in room temperature for 30 min, and a freshly prepared 40 mL aqueous solution of 0.02 mol/L sodium sulfide was added slowly. This suspension was stirred for 10 min and transferred to an autoclave. The autoclave was heated at 180 °C for 6 h to yield CNTs-Ag$_2$S. Thus, the formed CNTs-Ag$_2$S was collected by centrifugation and purified with washing with DI water.

### 3.3. Preparation of Ag$_2$S

For control experiments, Ag$_2$S nanoparticles were prepared using the procedures used for CNTs-Ag$_2$S without CNTs.

### 3.4. Adsorption Experiments

The stock solution of Cd(II), with a concentration of 1 g/L, was prepared in DI water using cadmium chloride and was diluted to the desired concentrations. The kinetic Cd(II) adsorption experiments were conducted to find the contact time needed to attain equilibrium. In the typical experiment, 100 mg of CNTs-Ag$_2$S was dispersed into 500 mL of a Cd(II) solution with a concentration of 0.5 mg/L and stirred at room temperature. After the required contact time, an adequate sample was collected, and the dispersed CNTs-Ag$_2$S was separated by centrifugation. The concentration of residual Cd(II) in the collected samples was estimated by atomic absorption spectrometer. The amount of Cd(II), adsorbed by CNTs-Ag$_2$S was monitored as a function of time for 120 min. The quantity of Cd(II) adsorbed was calculated using Equation (8).

$$q_t = \frac{(C_0 - C_t)\,V}{M} \tag{8}$$

where $q_t$ is the amount of adsorbed Cd(II) (mg/g) at time t; $C_0$ is the initial concentration of the Cd(II) (mg/L), and $C_t$ is the concentration of the Cd(II) (mg/L) at time t; V is the volume of the solution (L), and M is the amount of adsorbent (g).

Further, the efficiency of CNTs-Ag$_2$S in Cd(II) adsorption was estimated using Equation (9).

$$Removal\ efficiency\ (\%) = \frac{(C_0 - C_t)\,V}{C_0} \times 100 \tag{9}$$

For adsorption isotherm experiments, 10 mg of CNTs-Ag$_2$S was mixed with 50 mL of the Cd(II) solution and stirred at room temperature for 24 h to reach equilibrium in the concentration between 50 and 140 mg/L. After separating the dispersed CNTs-Ag$_2$S, the concentration of Cd(II) in the solution was measured using an atomic absorption spectrometer. The amount of Cd(II) adsorbed at equilibrium, $q_e$ (mg/g), was determined by Equation (10):

$$q_e = \frac{(C_0 - C_e)\,V}{M} \tag{10}$$

where $q_e$ is the amount of Cd(II) adsorbed (mg/g) at equilibrium.

### 3.5. Photocatalytic Activity

To evaluate the photocatalytic activity, 10 mg of CNTs-Ag$_2$S was added to the 100 mL aqueous solution of AYR with a concentration of 10 mg/L. It was allowed to stir in the dark for 30 min to reach adsorption/desorption equilibrium of the AYR molecules over the surface of CNTs-Ag$_2$S. This suspension was transferred to a photocatalytic reactor having a water jacket with a water circulation system to maintain a constant temperature, and the suspension was exposed to sunlight. At the required time, 5 mL of the reaction mixture was withdrawn, and the suspended CNTs-Ag$_2$S was separated using centrifugation. The concentration of AYR after photocatalysis was assessed with UV-vis spectrophotometer by recording the absorbance at 373 nm. The normalized concentration of AYR after photocatalysis was calculated as $C/C_0$, where $C_0$ is the initial concentration of AYR and C is its concentration after photocatalysis. All photocatalytic experiments were conducted in the month of June, between 1 pm and 4 pm. The intensity of sunlight measured during photocatalysis was 800–900 W/m$^2$.

### 3.6. Characterization

The UV-vis absorption spectra were recorded using a Jasco V-770 UV-vis-NIR spectrophotometer (Easton, MD, USA), and ATR-FTIR spectra were collected with a Smiths ChemID diamond attenuated total reflection (DATR) spectrometer (Smiths Detection, Inc., London, United Kingdom). The XRD was obtained by a Scintag X-ray diffractometer (Cupertino, CA, USA), model PAD X, equipped with a Cu-Kα photon source (45 kV, 40 mA), at a scanning rate of 3°/min. The thermogravimetry differential thermal analysis

(TG/DTA) was performed using a Perkin Elmer Diamond TG/DTA instrument (Waltham, MA, USA) at a 10 °C/min heating rate. Transmission electron microscopy (TEM) images and X-ray energy-dispersive spectroscopy (EDS) were perceived by a Hitachi H-8100 microscope (Tokyo, Japan). The X-ray photoelectron spectra (XPS) were acquired by a Perkin Elmer PHI 5600 ci X-ray photoelectron spectrometer (Waltham, MA, USA). The Cd(II) concentration was estimated using a Varian SpectrAA 220FS atomic absorption spectrometer (Lake Forest, CA, USA).

## 4. Conclusions

The facile hydrothermal process produced an efficient adsorbent and photocatalyst, CNTs-Ag2S. The ATR-FTIR, XRD, EDS, and XPS confirmed the formation of CNTs-Ag2S in the right structure and phase. The TEM explored the deposition of Ag2S nanoparticles over the surface of CNTs. The TG/DTA revealed the high thermal stability of CNTs-Ag2S. The dual-tasking CNTs-Ag2S could accomplish the complete Cd(II) adsorption and the degradation of AYR in water. The agreement of second-order kinetics for Cd(II) adsorption reveals that chemisorption is the rate-determining step of the adsorption process. The Weber−Morris intraparticle pore diffusion model represented that intraparticle diffusion could not be the sole rate-limiting step in Cd(II) adsorption, instead, it occurred through multiple phases. The validation of the Langmuir model illustrates that the Cd(II) adsorption takes place with monolayer molecular covering and chemisorption. Not limiting to Cd(II) adsorption, the CNTs-Ag2S could also be an excellent adsorbent for adsorption of other toxic heavy metals. Apart from excellent adsorbent, CNTs-Ag2S could also be an exceptional photocatalyst as reveled by degradation of AYR. The elevated degradation of AYR demonstrated that CNTs-Ag2S is the strong sunlight-active photocatalyst that could be applied in the degradation of other toxic dyes. The linearity of the L-H plots depicted that the degradation of AYR occurred through pseudo-first-order kinetics. The CNTs-Ag2S could be easily recovered and used for several times without losing its activity. Overall, CNTs-Ag2S is a robust adsorbent as well as photocatalyst that could be employed in the adsorption of different heavy metals and the photodegradation of alternative dyes.

**Supplementary Materials:** The following supporting information can be downloaded at: https://www.mdpi.com/article/10.3390/catal13030476/s1. Table S1: Parameters calculated for Cd(II) adsorption over CNTs-Ag2S from the intra-particle diffusion plot; Figure S1: EDS spectrum of CNTs-Ag2S; Figure S2: Plot perceived qt as a function of time for Cd(II) adsorption over CNTs- Ag2S; Figure S3: Intraparticle diffusion model for Cd(II) adsorption over CNTs- Ag2S; Figure S4: Freundlich isotherm plot for Cd(II) adsorption over CNTs- Ag2S; Figure S5: Temkin isotherm plot for Cd(II) adsorption over CNTs- Ag2S; Figure S6: Degradation of alizarin yellow R in presence of CNTs-Ag2S for successive three cycles under illumination to sunlight; Figure S7: XRD of CNTs- Ag2S (a) before and (b) after using in successive three cycles of photodegradation of alizarin yellow R under illumination to sunlight.

**Author Contributions:** G.M.N.: data curation, writing original draft, review, and editing funding acquisition. S.F.A.: investigation, data curation. E.A.J.: investigation, data curation. R.L.R.: writing, review, and editing. All authors have read and agreed to the published version of the manuscript.

**Funding:** The author (GMN) acknowledges the support of the National Academy of Sciences of the U.S.-Egypt Science and Technology Joint Fund and the Welch Foundation, Texas, United States, for departmental grant L-0002-20181021.

**Institutional Review Board Statement:** Not applicable.

**Informed Consent Statement:** Not applicable.

**Data Availability Statement:** Not applicable.

**Conflicts of Interest:** The authors declare no conflict of interest.

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
