# Peer review of "Adsorption Efficiency and Photocatalytic Activity of Silver Sulfide Nanoparticles Deposited on Carbon Nanotubes"

_catalysts, doi:10.3390/catal13030476_

Round 1

Reviewer 1 Report

Comments: Major revision is required.

1. FTIR spectra of all CNTs, Ag2S, CNTs–Ag2S should be discussed by showing the presence of proper functional groups present in the materials (Figure 1).

2. XRD patterns should be indexed with proper hkl values in figure 2.

3. Either low magnification TEM data of FESEM data should be provided for better understanding to the readers.

4. Exact weight loss should be shown in figure 4 with DTA to show the reaction nature (exothermic/endothermic).

5.  Full XPS survey should be indexed with proper binding energies corresponding to the elements present in the materials (Figure 6a).

6. Peak fitting is not good in high magnification XPS of the elements (Figure 6b, 6c, 6d, 6e).

7. Post catalysis characterization of the catalysts should be done for the stability.

8. What are the degraded products? Photo-degradation reaction mechanism has to be incorporated in text.

Author Response

Responses to reviewer’s comments

Manuscript ID: catalysts-2168452

Title: Adsorption efficiency and photocatalytic activity of silver sulfide
nanoparticles deposited on carbon nanotubes

We thank the reviewer for their valuable comments and recommendation to accept this manuscript for publication in Nanomaterials. We precisely addressed the reviewer’s comments by revising the manuscript accordingly. The outline below is details of our specific approaches and responses to criticism mentioned by the reviewer.

Reviewer-1

  1. FTIR spectra of all CNTs, Ag2S, CNTs–Ag2S should be discussed by showing the presence of proper functional groups present in the materials (Figure 1).

Response for 1: The FTIR spectra of CNTs-COOH (Figure 1a) and Ag2S (Figure 1b) have been added to the revised manuscript and discussed by comparing them with the spectrum CNTs–Ag2S.  

  1. XRD patterns should be indexed with proper hkl values in figure 2.

Response for 2: The reflection peaks in the XRD of CNTs–Ag2S have been marked in Figure 2.  

  1. Either low magnification TEM data of FESEM data should be provided for better understanding to the readers.

Response for 3: The low-magnification TEM images have been added to the revised manuscript and discussed (Figures 3e and f).

  1. Exact weight loss should be shown in figure 4 with DTA to show the reaction nature (exothermic/endothermic).

Response for 4: The percentage of weight loss from TGA has been mentioned. In addition, DTA curve has been added and discussed (Figures 4).    

  1. Full XPS survey should be indexed with proper binding energies corresponding to the elements present in the materials (Figure 6a).

Response for 5: The peaks in the survey spectrum of XPS (Figure 6a) have been marked.

  1. Peak fitting is not good in high magnification XPS of the elements (Figure 6b, 6c, 6d, 6e).

Response for 6: The XPS peaks have been fitted properly.

  1. Post catalysis characterization of the catalysts should be done for the stability.

Response for 7: The stability of CNTs–Ag2S was verified by recoding the XRD pattern after its application in three cycles of photodegradation of alizarin yellow R. The XRD pattern perused after photocatalysis showed no structure modification or rapture (Figure S7).       

  1. What are the degraded products? Photo-degradation reaction mechanism has to be incorporated in text.

Response for 8: The possible degradation products of photocatalysis were carbon dioxide, water, and other nontoxic compounds. The equations for the reaction paths of photocatalysis have been included in the revised manuscript.  

Reviewer 2 Report

In the paper entitled "Adsorption efficiency and photocatalytic activity of silver sulfide nanoparticles deposited on carbon nanotubes", Neelgund and co-workers described the preparation and evaluation of absorbing and photocatalytic properties of hybrid material consisting of CNTs and AgS NPs. The obtained results seem very interesting, and the paper is worth publishing in Catalysts. Several minor issues should be added/corrected:

1. In Fig. 5 and 6a, please mark the signals recorded with proper assignments for a better understanding of the description of spectra in the Results section.

2. In the case of heterogeneous photocatalytic material, it would be better to perform UV-Vis-DRS studies instead of classic UV-Vis spectra. If you have such a possibility, please perform UV-Vis-DRS.

3. Please compare the photocatalysis results of the photodegradation of the AYR with the literature data.

Author Response

Responses to reviewer’s comments

Manuscript ID: catalysts-2168452

Title: Adsorption efficiency and photocatalytic activity of silver sulfide
nanoparticles deposited on carbon nanotubes

We thank the reviewer for their valuable comments and recommendation to accept this manuscript for publication in Nanomaterials. We precisely addressed the reviewer’s comments by revising the manuscript accordingly. The outline below is details of our specific approaches and responses to criticism mentioned by the reviewer.

Reviewer-2

In the paper entitled "Adsorption efficiency and photocatalytic activity of silver sulfide nanoparticles deposited on carbon nanotubes", Neelgund and co-workers described the preparation and evaluation of absorbing and photocatalytic properties of hybrid material consisting of CNTs and AgS NPs. The obtained results seem very interesting, and the paper is worth publishing in Catalysts. Several minor issues should be added/corrected:

  1. In Fig. 5 and 6a, please mark the signals recorded with proper assignments for a better understanding of the description of spectra in the Results section.

Response for 1: The peaks have been marked in Figures 5 and 6(a).

  1. In the case of heterogeneous photocatalytic material, it would be better to perform UV-Vis-DRS studies instead of classic UV-Vis spectra. If you have such a possibility, please perform UV-Vis-DRS.

Response for 2: We do not have the facility of UV-Vis-DRS. We have to get it from outside. However, it is difficult to get the UV-Vis-DRS spectrum within the permitted time for submission of the revised manuscript, so we are unable to include this result in the revised manuscript. 

  1. Please compare the photocatalysis results of the photodegradation of the AYR with the literature data.

Response for 3: The photocatalytic activity of CNTs–Ag2S in the degradation of AYR has been compared with the reported photocatalysts and presented in Table 3. 

Round 2

Reviewer 1 Report

1. The main concern is still missing that is the photocatalytic degradation pathway. If possible, GC-MS or NMR techniques can be used to investigate the degraded products and will be helpful to show an appropriate reaction mechanism based on the degraded products. Author can follow the following work based on the degradation pathways published by others researchers, for examples;

https://www.sciencedirect.com/science/article/pii/S0926337300002769; https://www.sciencedirect.com/science/article/pii/S2214714422003099; https://pubs.rsc.org/en/content/articlelanding/2017/nj/c7nj02085f/unauth;

2. Low magnification TEM at the scale bar of 100 - 200 nm or FESEM should be provided to understand the exact morphology of the prepared materials.

Author Response

Responses to reviewer’s comments

Manuscript ID: catalysts-2168452

Title: Adsorption efficiency and photocatalytic activity of silver sulfide
nanoparticles deposited on carbon nanotubes

We thank the reviewer for their valuable comments and recommendation to accept this manuscript for publication in Nanomaterials. We precisely addressed the reviewer’s comments by revising the manuscript accordingly. The outline below is details of our specific approaches and responses to criticism mentioned by the reviewer.

Reviewer-1

  1. The main concern is still missing that is the photocatalytic degradation pathway. If possible, GC-MS or NMR techniques can be used to investigate the degraded products and will be helpful to show an appropriate reaction mechanism based on the degraded products. Author can follow the following work based on the degradation pathways published by others researchers, for examples;

https://www.sciencedirect.com/science/article/pii/S0926337300002769; 

https://www.sciencedirect.com/science/article/pii/S2214714422003099; 

https://pubs.rsc.org/en/content/articlelanding/2017/nj/c7nj02085f/unauth;

Response for 1: The possible reaction pathway has been represented by the following reactions. However, identifying the specific products formed by the photodegradation of Alizarin yellow R during photocatalysis like the above-referred manuscripts is a tedious process and needs advanced instrumentation and time. We do not have such facilities to study byproducts during photocatalysis and are not experts in organic reactions.      

  1. Low magnification TEM at the scale bar of 100 - 200 nm or FESEM should be provided to understand the exact morphology of the prepared materials.

Response for 2: The low magnification TEM at the scale bar of 200 nm has been added to Figure 3(a). 
